# REALISTIC EVALUATION OF TEST-TIME ADAPTATION: SURROGATE-BASED MODEL SELECTION STRATEGIES

## ABSTRACT

Test-Time Adaptation (TTA) has recently emerged as a promising strategy for tackling the problem of machine learning model robustness under distribution shifts. This setting constitutes a significant challenge as the model has to adapt to the new environment without any labeled data. Contemporary methods, such as neural networks, typically rely on a cumbersome hyper-parameter tuning procedure that leverages target labels, yet what happens when those labels are unavailable, as in the test-time adaptation scenario? In this work, we tackle this very problem of hyperparameter selection by evaluating several surrogate metrics (without any access to the test labels). The main goal of this work is to provide a realistic evaluation of TTA methods under different domain shifts, as well as evaluation of different strategies for model selection in TTA. Our main findings are: i) the accuracy of model selection strategies strongly varies across datasets and adaptation methods; ii) out of 6 evaluated approaches, only the AdaContrast method allows for surrogate-based model selection that matches oracle selection performance and iii) using a tiny-set of labeled test samples beats all competing selection strategies. Our findings underscore the need for future research in the field to conduct rigorous evaluations with explicitly stated model selection strategies, to give more realistic approximations of test-time adaptation methods performance.

## 1 INTRODUCTION

Machine learning models typically assume that training and testing data originate from similar distribution. However, in real-world applications, one observes a distribution shift between the training (source) and testing (target) data domains. That, in turn, hurts the performance (sometimes drastically) of the model during test time. Unsupervised domain adaptation methods explore adapting a model trained on the source domain to the target one, having access only to unlabelled target data. Test-time adaptation (TTA) extends this scenario to the *online* setting, where the model needs to adapt on the fly.

Recently, various approaches for test-time adaptation were developed. Some popular strategies involve entropy minimization on test data using pseudo-labeling (Wang et al., 2021). To increase the accuracy, extensive data augmentations can be used (Wang et al., 2022) or unreliable pseudo-labels can be filtered out (Niu et al., 2023). Yet, without any supervision at test time, errors in pseudo-labels are unavoidable. To avoid error accumulation model restart strategies are often employed (Niu et al., 2023; Wang et al., 2022), which are based on some heuristics. Despite, all of those challenges, existing test-time adaptation methods seem to work well, often significantly improving over the naive strategy (of using a source model without adaptation).

However, the reliability of such methods in practical applications remains dubious. Press et al. (2023) showed that under extremely long scenarios all existing TTA method results in degraded performance compared to the source model. Similarly, Zhao et al. (2023) demonstrated that existing TTA methods are very vulnerable to the choice of hyper-parameters. Interestingly, there is no consensus in the community on how to select the hyper-parameters for a given method. This is especially crucial for TTA where there is no access to the labelled data. The recent benchmark (Zhao et al., 2023) compared accuracies of various TTA methods assuming oracle selection strategy, considering access to the test data labels. This provides a fair comparison between existing methods, but can only give an overoptimistic measure of their potential performance. Furthermore, some methods

that score the best using oracle selection may still fall short when using a realistic model selection (e.g., due to the difficulty and sensitivity to hyper-parameter selection).

To this end, given an unlabeled dataset and a set of candidate models, we want to find an answer to the following research question: **how can we select the most accurate model**? We propose and evaluate different surrogate measures that do not use test labels for model selection in TTA. Our results give a realistic estimate of the existing TTA method's performance. A fair comparison with the oracle-based results can help with real-life TTA applications.

Our contributions can be summarized as follows:

- To the best of our knowledge, we are the first to provide an analysis of TTA methods performance under different model selection strategies. To this end, we identify intuitive and effective surrogate metrics of model performance and benchmark six state-of-the-art TTA methods, across four different datasets using six different model selection strategies.
- Our benchmark reveals that the accuracy of model selection strategies strongly varies across datasets and adaptation methods. In particular, we show that the ranking of existing TTA methods changes across different selection strategies (i.e., the best method under oracle selection strategy is never the best when using surrogate strategies). Additionally, we show that across evaluated TTA selection methods only AdaContrast allows for consistent model selection that matches oracle selection, without access to the target labels.
- With our detailed analysis we identify shortcomings of each of the analyzed surrogate metrics. We show that model selection is a very challenging problem, but access to the small labeled set greatly helps to improve the selection quality.

Our work emphasizes the importance of future research in this field to carry out thorough assessments, using clearly defined model selection strategies as benchmarks. We also create a testbed for a more rigorous comparison of existing TTA approaches.

## 2 RELATED WORK

**Model selection in TTA.**   Usually, no model selection strategy is provided (Niu et al., 2022; Gong et al., 2022; Niu et al., 2023; Wang et al., 2021; Döbler et al., 2023; Wang et al., 2022; Chen et al., 2022; Yuan et al., 2023), and only the final hyper-parameters are listed. This motivates us to perform rigorous evaluation of TTA methods, under different model selection strategies. A recent study shows that under oracle model selection strategy (accessing test labels) none of the existing methods are capable of addressing all common types of distribution shifts (Zhao et al., 2023).

There is only one non-oracle-based model selection strategy reported in the literature. In Rusak et al. (2022) authors use cross-dataset validation to perform hyper-parameters selection. In particular, they have used optimal parameters selected on the ImageNet-C dataset for testing on different benchmarks. We hypothesize that such a strategy can only work well if there is a similarity between the used datasets. Analogous strategy for parameter selection was used in (Boudiaf et al., 2022). The authors found that existing methods lack transferability, whereas the proposed parameter-free one (LAME) is robust to hyperparameter changes.

**Unsupervised model selection.**   Gulrajani & Lopez-Paz (2021) tested different strategies for unsupervised model selection in domain adaptation: i) source dataset accuracy ii) cross-dataset accuracy and iii) limited access to the target labels. They have shown that under such fair comparison, no domain adaptation method works better than standard empirical risk minimization training. This study was recently extended to partial domain adaptation by additionally incorporating other strategies based on surrogate (unsupervised) measures, such as entropy on model consistency on test data (Salvador et al., 2022).

The unsupervised model selection problem was also studied in time-series anomaly detection (Goswami et al., 2023). Based on different surrogate metrics authors proposed a robust rank aggregation method which shown to obtain competitive results. We are inspired by those studies and examine various strategies for test-time adaptation under different domain shifts, to i) obtain a realistic approximation of current methods' performance, and ii) validate surrogate measure applicability for hyper-parameter and model selection. One interesting aspect of test-time adaptation is

that the sequence length can have an impact on optimal model parameters, and this is something we also investigate by running test-time adaptation on ImageNet-C with different lengths.

# 3 TEST-TIME ADAPTATION PROBLEM

## 3.1 PROBLEM FORMULATION

Given the source model $f_{\theta_S}$, trained on the labeled data from the source domain $\mathcal{D}_S = \{\mathcal{X}_S, \mathcal{Y}_S\}$, the objective is to adapt it to the test data $\mathcal{D}_T = \{\mathcal{X}_T, \mathcal{Y}_T\}$. In the source data $\mathcal{D}_S$, sample-label pairs $(x_i \in \mathcal{X}_S, y_i \in \mathcal{Y}_S)$ are distributed according to the probability distribution $\mathcal{P}_S(\boldsymbol{x}, y)$. The test samples $x_i \in \mathcal{X}_T$ are modeled by a different distribution $\mathcal{P}_T(\boldsymbol{x}, y) \neq \mathcal{P}_S(\boldsymbol{x}, y)$. At each testing time step $t$, the TTA process aims to adapt the model $f_{\theta_S}$ to better align with the distribution $\mathcal{P}_T$, Importantly, unlabeled data samples $\boldsymbol{x}^t \sim \mathcal{P}_T$ are the only information available regarding the unseen distribution $\mathcal{P}_T$.

Model selection, that is selecting both (i) an algorithm and (ii) its associated hyperparameter(s) is a crucial part of all machine learning pipelines. In a standard supervised learning setting, a validation set can be used to estimate the model's accuracy. However, in TTA such an approach is not possible as there is no access to labeled target samples. Below, we discuss several approaches for model selection in TTA which we use in this work.

## 3.2 MODEL SELECTION STRATEGIES IN TTA

**Source accuracy** (S-ACC). The most simple strategy uses a small validation set from the source domain to estimate the model's performance in the target domain, e.g., in domain adaptation (Ganin & Lempitsky, 2015). This strategy assumes that the training and test examples follow similar distributions. However, interestingly, Miller et al. (2021) shows that there is a linear correlation between accuracy on i.i.d. and o.o.d. data which may favor such a strategy.

Some TTA methods assume there is no access to the source data, e.g., because of privacy concerns (Liang et al., 2020). In such applications, this strategy may not be valid.

**Cross-validation accuracy** (CROSS-ACC). Another reasonable approach is to select hyperparameters on some other dataset. This is a potentially valid strategy since already plenty of benchmarks exist for TTA, which could be utilized for this purpose. Here, we follow Rusak et al. (2022) and perform hyper-parameter selection on ImageNet-C dataset, which they show to work well. In our work, we additionally test whether the parameters tuned on one dataset also work well when we test on the same dataset i) but with increased sequence length, and ii) when temporal correlation between classes is changed.

**Entropy** (ENT). Minimizing the Shannon entropy $H(\hat{y}) = -\sum_c p(\hat{y}_c) \log p(\hat{y}_c)$, where $\hat{y}_c$ is a probability prediction of class $c$, of predictions for target samples $\hat{y} = f_\theta(\boldsymbol{x}^t)$ was introduced for test-time adaptation in Wang et al. (2021) because i) it is related to error, as more confident predictions are all-in-all more correct and ii) entropy is related to shifts due to corruption, as more corruption results in more entropy. This strategy was adopted by several works showing its efficiency (Niu et al., 2023; Wang et al., 2021; Niu et al., 2022). However, as shown by Boudiaf et al. (2022) in some scenarios the entropy does not correlate with the accuracy. Nevertheless, in our work, we decided to also include entropy as the surrogate measure, because of its simplicity and common usage.

**Model consistency** (CON). Consistency regularization is an important component of many self-supervised learning algorithms (Sohn et al., 2020). It enforces the model $f_\theta$ to have similar predictions when fed perturbed versions $\tilde{\boldsymbol{x}}_i$ of the same image $\boldsymbol{x}_i$ ($f_\theta(\boldsymbol{x}_i) \simeq f_\theta(\tilde{\boldsymbol{x}}_i)$). It was also used to drive training of the model during the TTA phase (Nguyen et al., 2023). As the choice of perturbations, we use the augmentation pipeline introduced in CoTTA (Wang et al., 2022), as it is quite commonly used in TTA and was shown to work well. The augmentations include color jittering, Gaussian noise, and blur, etc., further information can be found in the Appendix (sec. A).

**Using test labels.** We also incorporate ORACLE model selection strategy to measure upper-bound for evaluated methods and to see how close to it the other surrogate measure performs. Additionally, we use one more additional model selection strategy, 100-RND, that assumes limited access to the

| TTA Methods | TENT, SAR, LAME, EATA, RMT, ADACONTRAST |
|---|---|
| Model Selection Strategies | S-ACC, ENT, CROSS-ACC, DEV, 100-RND, ORACLE |
| Tuned parameters | learning rate, momentum and method-specific parameters (SAR, EATA) |
| Experimental protocol | 4 datasets, varying levels of temporal correlation and adaptation sequence length |

Table 1: Summary of experimental setup considered in this work.

test data. Following Salvador et al. (2022) we use labels from 100 randomly selected images from test data for model selection.

## 4 EXPERIMENTAL SETUP

### 4.1 DATASETS

For experiments, we utilize two widely used datasets for TTA evaluation with artificial image corruptions: CIFAR100-C and ImageNet-C (Hendrycks & Dietterich, 2019). Moreover, we use two datasets consisting of images without corruptions, but portraying objects in different domains: DomainNet-126 (Saito et al., 2019) and ImageNet-R (Hendrycks et al., 2021).

CIFAR100-C and ImageNet-C datasets are created by applying 15 different corruption types with 5 severity levels to the images from the test split of the clean CIFAR100 (Krizhevsky, 2009) and ImageNet (Deng et al., 2009) datasets, creating multiple domains. Following the state-of-the-art TTA works (Wang et al., 2022; Niu et al., 2022; Döbler et al., 2023), for testing TTA methods we utilize the standard sequence of corruption types with the 5th level of severity sequentially, without mixing the domains. The source models are trained on the clean CIFAR100 or ImageNet dataset.

DomainNet-126 is a cleaned from noisy labels subset of the DomainNet dataset (Peng et al., 2019). It consists of images of 126 different objects from 4 different domains: real-world photos, clipart images, sketches, and paintings. The source model is trained on actual photos and adapted during test time to the images from the sequence of the remaining domains.

ImageNet-R is composed of various renditions of 200 objects from the original ImageNet dataset, spanning various domains such as art, cartoons, graffiti, embroidery, origami, tattoos, and more. This dataset comprises a total of 30000 images. Source data come from the original ImageNet and TTA is tested on all of the rendition types at once.

Additionally, inspired by Gong et al. (2022) we test the influence of temporally correlated class distribution within the CIFAR100-C test sequence on the performance of different models of TTA methods. For this purpose, we utilize Dirichlet distribution with different concentration parameters $\delta$ to sample the class exemplars. Moreover, we test the scenario of a long TTA operation by utilizing the ImageNet-C testing sequence ten times without any model reset in between the sequences.

### 4.2 METHODS

In our testbed, we include the six different methods, which differ by the optimized objective, number of hyper-parameters, fine-tuned layers etc. TENT (Wang et al., 2021) is a popular TTA method that includes minimizing the entropy of the predictions while optimizing only batch norm statistics of the model. SAR (Niu et al., 2023) additionally applies filtering to remove noisy samples, uses a sharpness-aware optimizer that encourages flat minima, and finally applies a model reset scheme. EATA (Niu et al., 2022) uses a filtering scheme that removes not only noisy samples but also redundant ones and additionally introduces a Fisher regularizer to constrain important model parameters from drastic changes (this requires one-time access to the source data). In contrast to previous methods that use entropy minimization objective and fine-tune only batch normalization layers, AdaContrast (Chen et al., 2022) incorporates self-supervised contrastive objective with consistency regularization and fine-tune all weights of the model. RMT method (Döbler et al., 2023) also uses contrastive objective but additionally uses it to align the learned feature space with one of the source model combined with introduced symmetrical cross-entropy loss and weight averaging.

**Hyper-parameter sensitivity.** A potential limitation of the above methods is that they require per-task parameter tuning, which can be a serious limitation in test-time adaptation, without access to test labels. In LAME, Boudiaf et al. (2022) analyzed hyperparameters sensitivity of the existing TTA methods and proposed a parameter-free approach that modifies only the model's output (not its parameters) and solving the objective with a concave-convex procedure (i.e., without the need to tune learning rate). As a result, per each dataset, we run only one test with the LAME method and do not need to perform the hyper-parameter selection.

**Access to the source data.** RMT method additionally assumes a replay mechanism of the source data, which may give this method an advantage over other methods, which were developed with the motivation of not using a memory buffer during training. Thus, we experiment with RMT source-free variant which we denote as RMT-SF. It was shown in the original paper to also work well.

### 4.3 EXPERIMENTAL SETUP

**Hyper-parameters.** For each of the algorithms we perform a search over the two most important hyperparameters, that is: the learning rate (4 configurations) and momentum (using a value of 0.9 and not using momentum at all), similar to (Boudiaf et al., 2022). This gives 8 configurations per model. For SAR and EATA we additionally search over the method-specific parameters using random search, which gives 8 additional configurations per those methods. Details about the hyper-parameters are given in the Appendix (sec. A). Each experiment is repeated 3 times. In total, we performed 1500 adaptation sessions. All models are compared against the SOURCE model, which assumes no adaptation at all.

**Implementation details.** For CIFAR100-to-CIFAR100-C following the RobustBench benchmark (Croce et al., 2021) ResNeXt-29 (Xie et al., 2017) architecture is used. For remaining datasets, a source pre-trained ResNet-50 is used similar to Döbler et al. (2023). As for the optimizer we use SGD for all the methods. For the choice of batch size, we are the most interested in the *online* setting, that is batch size equal one. However, some methods do not work well in the single-image adaptation setting. As such, we use batch size equal to ten, which was shown in other works to work still well (Niu et al., 2023). A batch size equal to ten is also commonly used in the continual learning for *online* scenarios (onl, 2022).

**Test-time augmentations**. Some of the TTA methods use data augmentation at test-time (Wang et al., 2022; Sun et al., 2020), including AdaContrast and RMT considered in this paper. Those augmentations include some of those that were used to generate ImageNet-C and CIFAR100-C benchmarks (details in the Appendix). We treat those augmentations as part of the method as they are needed to compute consistency loss, however, it is important to note that those augmentations can give an advantage for those methods on the first two benchmarks. In general, knowledge of a distortion type that can occur at test time can be beneficial and could be used when designing a TTA method.

### 5 RESULTS

In Table 2 we show the results of hyperparameter selection under different selection strategies. In Fig. 1 we plot correlations between surrogate metrics and target accuracy. Further, we increase the difficulty of the testing conditions by increasing the length of testing scenarios in Table 4 and by varying the class correlation level of the test samples in Fig. 3.

Finally, we extend our testing scenario to the model selection strategy in Table 3 to show the performance of surrogate metrics when selecting across different TTA methods.

**Varying gap between surrogate-based hyper-parameter selection and oracle selection**. The gap varies between different methods. For example, for the EATA method when using a source selection strategy the gap varies from 0.66% on CIFAR100-C to 7.11% on ImageNet-C (which accounts for 24.62% of the relative gap). On the contrary, for the AdaContrast method using source accuracy for model selection allowed to match the oracle selection strategy on all of the datasets.

**Different methods ranking under different evaluation strategies**. While EATA is significantly the best under the oracle selection strategy (49.99 on average) it is outperformed for example by Tent

| DATASET | METHOD | S-ACC | CROSS-ACC | ENT | CON | 100-RND | MEDIAN | ORACLE |
|---|---|---|---|---|---|---|---|---|
| CIFAR100-C | SOURCE | | | | 53.55 | | | |
| | TENT | 59.93±0.15 | 59.56±0.2 | 5.32±0.05 | 59.56±0.20 | 59.17±0.41 | 54.46±0.51 | 60.16±0.14 |
| | EATA | 61.78±0.40 | 61.86±0.47 | 21.01±20.12 | 58.60±4.71 | 61.95±0.57 | 60.51±0.36 | 62.44±0.35 |
| | SAR | 59.47±0.11 | 45.65±5.83 | 11.68±5.92 | 48.75±0.21 | 59.26±0.2 | 57.03±0.27 | 60.01±0.12 |
| | ADACONTRAST | 64.43±0.01 | 63.70±0.11 | 63.69±0.11 | 37.97±0.43 | 64.43±0.01 | 48.48±0.56 | 64.43±0.01 |
| | LAME | | | | 53.014±0.03 | | | |
| | RMT-SF | 59.22±0.13 | 60.04±0.1 | 60.17±0.07 | 60.17±0.07 | 59.93±0.13 | 60.06±0.12 | 60.33±0.11 |
| IMAGENET-C | SOURCE | | | | 17.97 | | | |
| | TENT | 27.98±0.07 | 31.08±0.06 | 1.52±0.1 | 29.79±0.16 | 30.29±0.76 | 28.83±0.03 | 31.37±0.03 |
| | EATA | 28.87±0.11 | 14.60±12.38 | 1.96±2.28 | 23.39±16.54 | 32.25±0.99 | 18.74±8.45 | 35.98±1.02 |
| | SAR | 27.14±0.09 | 28.72±0.11 | 8.41±4.6 | 31.19±0.22 | 30.93±0.35 | 29.30±0.45 | 31.30±0.08 |
| | ADACONTRAST | 33.29±0.09 | 31.81±0.14 | 33.29±0.09 | 3.90±0.26 | 31.81±2.01 | 18.94±0.13 | 33.28±0.09 |
| | LAME | | | | 17.72±0.02 | | | |
| | RMT-SF | 28.81±0.05 | 27.54±2.6 | 23.76±0.1 | 23.76±0.1 | 30.51±0.1 | 29.16±0.13 | 31.54±0.15 |
| DOMAINNET-126 | SOURCE | | | | 54.71 | | | |
| | TENT | 49.92±0.08 | 52.13±0.06 | 8.21±0.11 | 52.13±0.06 | 52.04±0.19 | 50.32±0.08 | 52.24±0.01 |
| | EATA | 50.17±0.06 | 53.37±0.92 | 19.76±23.79 | 53.59±0.95 | 53.30±0.54 | 51.88±0.19 | 54.44±0.27 |
| | SAR | 49.73±0.09 | 50.91±1.15 | 28.48±5.55 | 51.24±0.19 | 51.45±0.44 | 50.55±0.31 | 52.75±0.07 |
| | ADACONTRAST | 56.51±0.04 | 55.69±0.15 | 46.50±0.42 | 19.61±0.79 | 54.32±3.11 | 51.00±0.79 | 56.51±0.04 |
| | LAME | | | | 54.34±0.01 | | | |
| | RMT-SF | 51.02±0.05 | 52.67±0.02 | 46.36±0.64 | 46.36±0.64 | 52.40±0.20 | 52.42±0.11 | 52.70±0.01 |
| IMAGENET-R | SOURCE | | | | 36.17 | | | |
| | TENT | 37.43±0.28 | 38.92±0.07 | 11.01±0.59 | 38.30±0.17 | 38.40±0.13 | 37.55±0.14 | 38.92±0.07 |
| | EATA | 37.17±0.13 | 43.19±0.75 | 2.35±1.46 | 42.27±0.18 | 42.03±0.43 | 39.70±0.83 | 43.09±0.76 |
| | SAR | 37.09±0.73 | 41.94±0.24 | 24.28±0.71 | 41.94±0.23 | 41.31±1.06 | 37.84±0.17 | 41.94±0.24 |
| | ADACONTRAST | 39.23±0.12 | 39.53±0.14 | 37.94±0.26 | 15.00±0.39 | 38.86±0.78 | 35.80±0.18 | 39.53±0.14 |
| | LAME | | | | 35.95±0.01 | | | |
| | RMT-SF | 36.81±0.16 | 40.97±0.32 | 40.97±0.32 | 40.97±0.32 | 40.64±0.78 | 38.77±0.10 | 40.97±0.32 |
| AVERAGE | TENT | 43.81-(4) | 45.42-(2) | 6.51-(6) | **44.95-(1)** | 44.98-(5) | 42.79-(3) | 45.67-(5) |
| | EATA | 44.50-(2) | 43.13-(4) | 11.27-(5) | 44.47-(2) | **47.38-(1)** | 42.71-(4) | **49.99-(1)** |
| | SAR | 43.36-(5) | 41.81-(5) | 18.22-(4) | 43.28-(3) | 45.74-(4) | 43.68-(2) | 46.50-(3) |
| | ADACONTRAST | **48.37-(1)** | **47.68-(1)** | **45.36-(1)** | 19.12-(6) | 47.36-(2) | 38.56-(6) | 48.44-(2) |
| | LAME | 40.26-(6) | 40.26-(6) | 40.26-(3) | 40.26-(5) | 40.26-(6) | 40.26-(5) | 40.26-(6) |
| | RMT-SF | 43.96-(3) | 45.31-(3) | 42.82-(2) | 42.82-(4) | 45.87-(3) | **45.10-(1)** | 46.39-(4) |

Table 2: Final accuracy of evaluated test-time adaptation methods under different model selection strategies. Green color marks the best surrogate strategy for each of the methods, and the red color marks the worst one. The last section aggregates the results over the 4 datasets, here we rank each of the TTA methods within each of the selection strategies. **Bolded** text marks the best one, and the underline marks the second best method.

| DATASET | S-ACC | CROSS-ACC | ENT | CON | 100-RND | MEDIAN | ORACLE |
|---|---|---|---|---|---|---|---|
| CIFAR100-C | 53.01±0.03 | 61.87±0.47 | 3.91±2.04 | 60.17±0.06 | 62.08±1.78 | 59.17±0.18 | 64.44±0.01 |
| IMAGENET-C | 17.7±0.02 | 31.81±0.14 | 1.08±0.67 | 3.90±0.26 | 32.15±1.11 | 28.75±0.17 | 35.98±1.02 |
| DOMAINNET-126 | 54.35±0.01 | 53.25±0.38 | 31.61±20.7 | 19.61±0.79 | 53.02±0.98 | 51.25±0.07 | 56.51±0.04 |
| IMAGENET-R | 35.95±0.01 | 43.09±0.76 | 27.54±18.8 | 14.99±0.39 | 41.91±0.56 | 38.14±0.23 | 43.09±0.77 |

Table 3: Target accuracy for different model selection strategies when selecting from a pool of all trained models (57 models in total). When selecting the checkpoint across different methods, the model selection is more difficult and the gap between oracle and other selection methods increases. Using the source-accuracy strategy resulted in a very conservative model selection of models that did not adapt much.

(5th method using oracle selection) when using 3 out of 4 surrogate-based metrics. Notably, TENT achieves the 2nd best score across all methods when using surrogate-based metrics (mean accuracy of 45.42). AdaContrast is clearly the best method when using surrogate-based metrics scoring as the 1st method on all metrics except for consistency loss.

**On usage of metrics that were part of the training**. Using entropy to drive model selection performs poorly if it is one of the main training objectives (SAR, TENT, EATA). This seems intuitive as optimizing for the entropy using pseudo-labels might lead to degenerate solutions, e.g., consistently predicting only one class with high confidence. This also holds true for the consistency loss and the AdaContrast method. We take a closer look at this phenomenon and see that initially optimizing entropy leads to better selection results (Fig. 1 first row and column). Nevertheless, once a certain threshold is reached, the model's generated solutions gradually deteriorate in quality. We could select a "reasonable" range of entropy by computing it on the source dataset, but this would still need

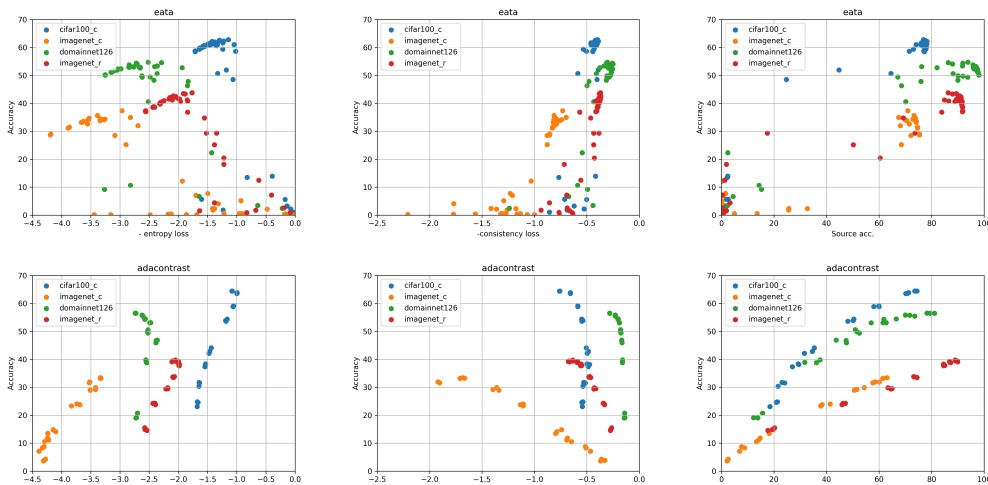

Figure 1: Correlation plots for the EATA (upper row) and AdaContrast (bottom row) method between target accuracy (y-axis) and selected surrogate measures (x-axes). **EATA**: (Left) Selecting the model based on the entropy loss works up to some point, after which minimizing the entropy leads to degenerate solutions with small accuracy. However, there is a smooth transition between favorable to unfavorable solutions exists and it requires manual threshold selection. (Center) Selecting the model based on the consistency loss leads to solutions close to the optimal, however, the margin between viable and suboptimal solutions is rather small. (Right) Selecting the model based on the source accuracy leads to selecting fairly good solutions and it clearly isolates degenerated models. **AdaContrast**: Contrary to the EATA, entropy-based model selection works quite well, and the consistency-based selection produced degenerated solutions. Using source data to measure accuracy allows for optimal model selection.

tuning the parameters, which is something we wanted to avoid. Interestingly, for the RMT method which uses a multi-task loss combined with access to the source prototypes, model selection using entropy or consistency works reasonably well, usually a few percent behind other surrogate-based selection strategies.

**No single best surrogate metric**. Our results show that there is no surrogate metric that consistently selects the best model on all of the datasets and methods (Fig. 2). The two most promising strategies are based on source accuracy and using a cross-validation selection mechanism. However, there are some exceptions. For example, EATA scored 14.60 on the ImageNet-C benchmark (compared to the 35.98 of oracle selection) using the CROSS-ACC strategy. This might be due to the fact of the rather large number of hyperparameters of the EATA method and changes in dataset characteristics. Using source accuracy for the selection mechanism produces stable results, but is often far from optimal. For example, for the TENT method, using source accuracy for the selection strategy generates results that are better only from the entropy selection method (see Table 5 for ranking of selection strategies). One potential caveat of using a source-accuracy selection strategy is that it may result in "conservative" model selection, that is models that do not change much from the source model.

**Gap significantly increases when selecting across different methods**. When performing model selection across a pool of all trained models (across different methods), Table 3, the gap significantly increases. Notably, when selecting the model based on the source accuracy in all of the cases the LAME method was selected as it is a conservative method that does not change much its outputs. As a result, using this strategy an accuracy of 17.7 was achieved on ImageNet-C benchmark, compared to the 35.98 of oracle selection. This shows that each of the surrogate-based strategies has its own limitations, and probably should not be used in isolation.

**Limited access to the target labels helps a lot.** As shown in the results in Table 2, having access to just one hundred target labels helps to consistently improve over the other surrogate-based model selection strategies. In most of the cases, this strategy results in selecting hyper-parameters that are within 1% of accuracy compared to the oracle selection.

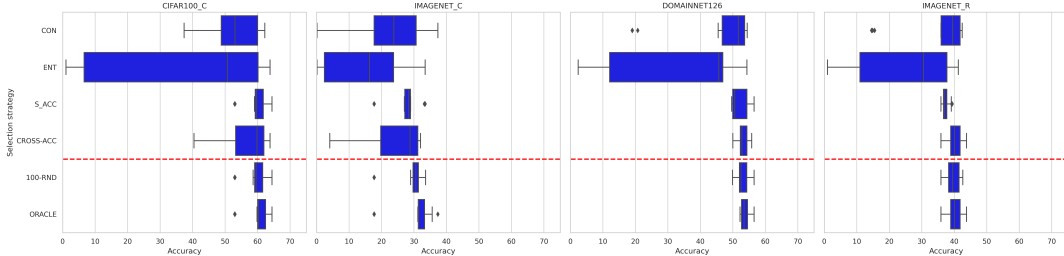

Figure 2: Target accuracy per selection strategy for different datasets, results aggregate all evaluated test-time adaptation methods. Using test labels (Oracle and 100-rnd) performs consistently the best. Using the Cross-Acc strategy works really well on ImageNet-R, but not so well on other datasets with evident outliers. The horizontal line separates selection strategies that use target labels.

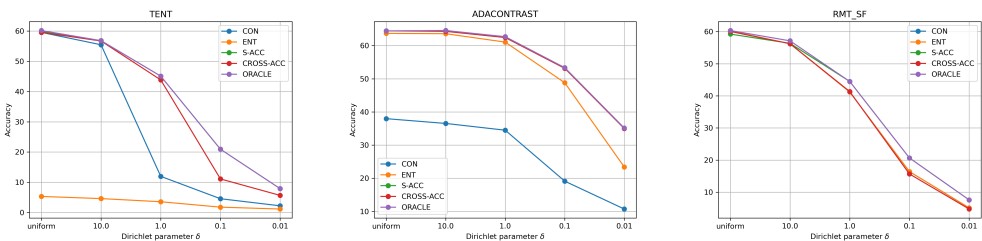

Figure 3: CIFAR100-C results: the effect of Dirichlet concentration parameter $\delta$ on the model performance for different methods and selection strategies. Even if the surrogate-based metrics match oracle accuracy in the base scenario (uniform class distribution) they may fall short when the temporal correlation is changed.

**TTA remains very challenging.** In Table 4 we presents results for the Long ImageNet-C scenario. Here, when using cross-acc metric we have used oracle parameters obtained on the original ImageNet-C scenario. We show that it performed poorly for 2 out of 4 evaluated methods (TENT and SAR). This shows that the optimal parameters depend on the length of the sequence (with some methods being robust to such changes). What's more, in Fig. 3 we controlled for the temporal classes correlation. As presented in the figures the difference between surrogate-based model selection and oracle one increases for some of the selection strategies as we increase task difficulty.

**Performance of TTA methods.** In our experiments we use two very popular benchmarks for TTA (CIFAR100-C and ImageNet-C) and two less popular ones (DomainNet-126 and ImageNet-R). As expected, all the methods easily beat the source model in the first two settings. However, when testing on ImageNet-R most of the methods improve over the source model but with a minor margin, and on DomainNet-126 AdaContrast method is the only one that does not degrade the performance. The lack of success on the last two datasets is due to the fact that they are not commonly used and we apply a small batch size (RMT method also showed its performance when using batch size = 1, but they have used exemplars in that setting). This overall underscores the fact that the TTA remains challenging, with questionable transferability to novel testing conditions.

## 6 DISCUSSION

**Offline vs. online selection strategy**. In this work, we have considered an offline model selection strategy, that is we run adaptation using multiple models and select the best model afterward using some measures. In TTA one we would ideally select the parameters on the fly by adjusting to the testing conditions. Nevertheless, evaluating whether existing surrogate measures can serve the purpose of estimating model performance on target data is the first step in that direction.

**On consistency of stability of surrogate-measures.** In general, we have found that existing surrogate-based metric, after removing the outliers (so for example discarding entropy-based se-

| DATASET | METHOD | S-ACC | CROSS-ACC | ENT | CON | MEDIAN | ORACLE |
|---------|--------|-------|-----------|-----|-----|--------|--------|
| IMAGENET-C LONG | TENT | 29.20±1.8 | 25.62±8.11 | 0.65±0.56 | 30.51±0.86 | 22.70±5.3 | 31.48±0.17 |
| | EATA | 30.23±1.99 | 34.64±0.54 | 0.41±0.3 | 22.49±15.82 | 18.81±6.88 | 35.2±0.41 |
| | SAR | 28.57±1.19 | 25.35±4.17 | 10.34±9.13 | 25.78±3.55 | 29.44±0.65 | 32.18±1.43 |
| | RMT-SF | 28.79±0.06 | 31.53±0.15 | 25.48±0.23 | 25.48±0.23 | 29.56±0.19 | 31.52±0.15 |

Table 4: Target accuracy for different hyper-parameter selection strategies when selecting for a Long ImageNet-C scenario. Using oracle parameters from standard ImageNet-C benchmark (Cross-Acc strategy) performs poorly for SAR and EATA. Also, greater variance in results can be observed.

lection for SAR, EATA, TENT methods), works reasonably well, usually within a few percent of accuracy to the oracle selection, on average. However, all the metrics lack the required stability, and for each of those we have identified at least one scenario when it worked very poorly.

**Need for more real-world benchmarks**. Despite the best efforts the benchmark proposed here (and in other works) is only a very simplified version of real-world challenges and the conclusions drawn from this research should be applied very carefully to the real world. In particular, we envision that the challenge of using TTA methods over very long (potentially infinitely long) adaptation phases still requires much more research in this direction.

# 7 CONCLUSIONS

In this work we evaluated existing test-time adaptation methods on a diverse array of benchmarks using practical model selection strategies, that is without accessing target labels. Throughout the experiments, we found numerous findings. We showed that the accuracy of model selection strategies strongly varies across datasets and adaptation methods and out of all evaluated approaches, only the AdaContrast method allows for surrogate-based model selection that matches oracle selection performance. Finally, we showed the limitations of the current surrogate-based metric, which are all inferior to a strategy of using a tiny set of labeled test samples. The study has important implications from a practical point of view and shows the importance of the model selection problem in TTA, which we believe did not receive enough attention from the community.

In the experiments, we used various datasets, as well as we controlled for the length of testing sequences and their correlation ratio, imitating real-world conditions. Yet, those provide only a simplification of challenges occurring in the real world and therefore we believe that more work is required in the construction of relevant TTA benchmarks. However, we believe that our work is an important step toward harnessing the potential of TTA methods.

**Reproducibility statement.** We provided all the necessary information required to repeat the experiments. In Sec. 4 (main paper) and Sec. A (Appendix), we refer to the repository we have used to provide all the implementation details, hyper-parameter configurations etc., and the code itself. We will open-source our code upon acceptance.

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

This document describes our experimental procedure in detail and provides additional plots and figures. The code is provided in the supplementary file.

## A  EXPERIMENTAL SETUP

**Hyper-parameters.** We use the following hyper-parameter search:

- Learning rate $\in \{(0.001, 0.00025, 0.0000625, 0.0000156)\}$
- Momentum $\in \{0.0, 0.9\}$
- For SAR method:
  - The parameter $E_0 \in \{0.1, 0.4, 0.8\}$
  - Reset threshold $e_0 \in 0.01, 0.05, 0.2, 0.5$
- For EATA method:
  - The parameter $E_0 \in \{0.1, 0.4, 0.8\}$
  - $\epsilon$ hyperparameter for similarity thresholding $\in \{0.05, 0.1, 0.4, 0.8\}$
  - $\beta$ parameter for Fisher regularization $\in \{1, 1000, 2000, 5000\}$
- Access to source data:
  - EATA - we use 2000 source samples to compute Fisher regularization term.
  - RMT - here we use 10% of the original source data to compute source prototypes.

**Data Augmentations.** Our calculation of model consistency includes the same augmentations as in CoTTA paper Wang et al. (2022), which is a combination of color jittering, gaussian blur, gaussian noise, affine transformations, and image cropping. The same augmentations are performed by the RMT method during the adaptation phase.

AdaContrast method used during adaptation phase augmentations from the MoCo method (He et al., 2020) used in representation learning. The augmentations include image resizing, color jittering, grayscale augmentation, gaussian blur, and horizontal flip. For details about CoTTA and MoCo augmentation please refer to the repository that we have used: https://github.com/mariodoebler/test-time-adaptation.

## B  ADDITIONAL RESULTS

Table 5 presents results of model selection strategies for different adaptation methods, but for convenience, here we rank the TTA methods (instead of model selection strategies as in Table 2.

Fig. 4 show correlation plots for different adaptation methods, not included in the main paper (Fig.1).

In Fig. 5 we show the accuracy of all evaluated configurations of test-time adaptation for our main datasets.

| METHOD | S-ACC | CROSS-ACC | ENT | CON | 100-RND | ORACLE |
|---|---|---|---|---|---|---|
| TENT | 43.81-(5) | 45.42-(2) | 6.51-(6) | 44.95-(4) | 44.98-(3) | 45.67-(1) |
| EATA | 44.50-(3) | 43.13-(5) | 11.27-(6) | 44.47-(4) | 47.38-(2) | 49.99-(1) |
| SAR | 43.36-(3) | 41.81-(5) | 18.22-(6) | 43.28-(4) | 45.74-(2) | 46.50-(1) |
| ADACONTRAST | 48.37-(2) | 47.68-(3) | 45.36-(5) | 19.12-(6) | 47.36-(4) | 48.44-(1) |
| RMT-SF | 43.96-(4) | 45.31-(3) | 42.82-(5) | 42.82-(5) | 45.87-(2) | 46.39-(1) |

Table 5: Model selection strategies ranking for each of the evaluated methods, results aggregated across 4 datasets (CIFAR100-C, ImageNet-C, DomainNet-126, ImageNet-R).

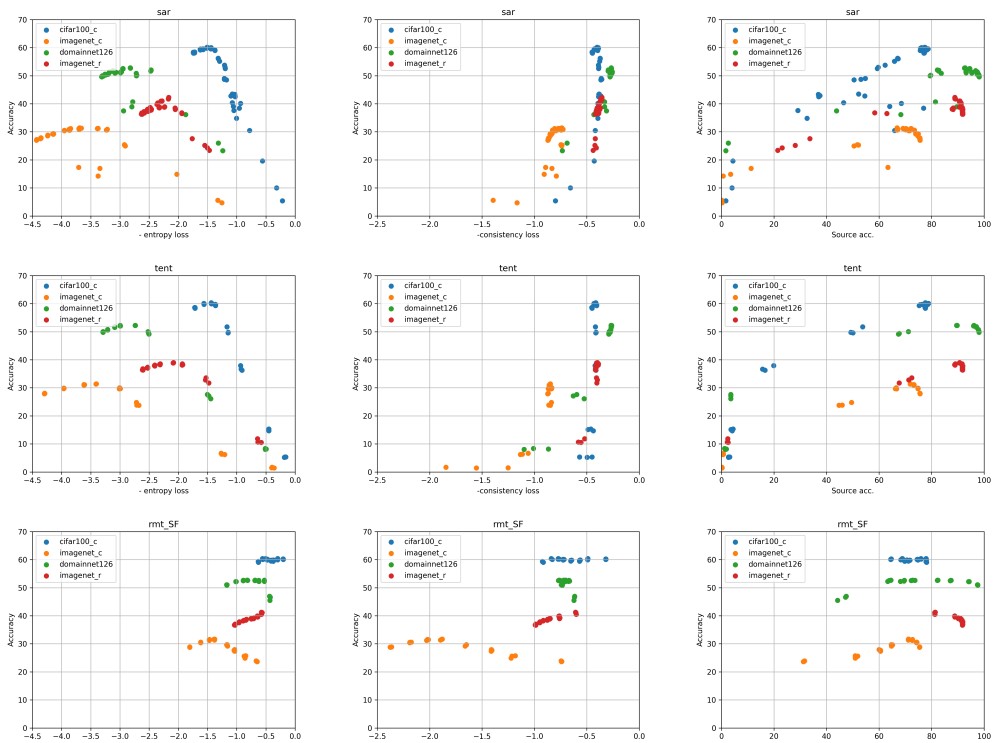

Figure 4: Correlation plots for the different methods between target accuracy (y-axis) and selected surrogate measure (x-axis).

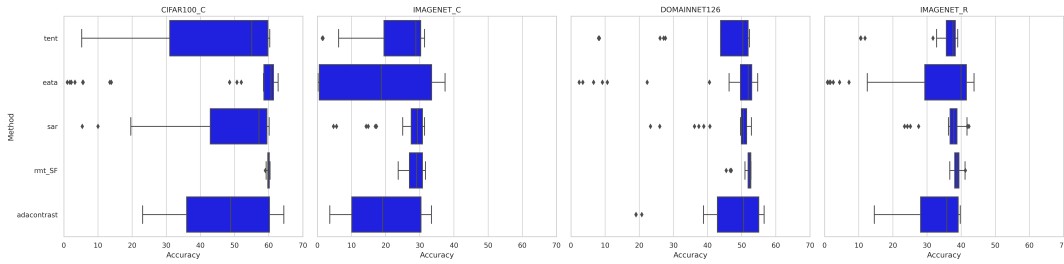

Figure 5: Target accuracy per TTA methods for different datasets, results aggregate all evaluated hyper-parameter configurations.

