# OpenReview forum: "Realistic Evaluation of Test-Time Adaptation: Surrogate-Based Model Selection Strategies"
_ICLR.cc/2024/Conference — ICLR 2024 Conference Withdrawn Submission_

### Official Review · Reviewer_Yunj · 2023-10-24

**Soundness:** 3 good
**Presentation:** 2 fair
**Contribution:** 1 poor
**Rating:** 3
**Confidence:** 4

**Summary:**

This paper presents an empirical analysis on cross validating Test Time Adaptation (TTA) methods using surrogate unsupervised metrics. By doing so, one could unlock the ultimate potential of each TTA method and choose its best hyper-parameters. The main findings of this work show that the most current surrogate unsupervised loss functions used in TTA are incapable of hinting about the best TTA method (best hyper parameters choices) and validating using a tiny set of labeled examples can outperform them. Further, such loss functions are also sensitive to the length of the stream of data presented to the adaptation method.

**Strengths:**

The main strengths of this work are:

- The problem this work is studying is important. Picking the best adaptation method (with its best hyper parameters) will allow better and more reliable real-world deployment of pre-trained models when exposed to distribution shifts.

- The methodology of this paper incorporated 5 different approaches for model selection.

- Experiments included multiple TTA benchmarks (4 in total).

- Experiments included 6 different TTA methods.

**Weaknesses:**

The main weaknesses of this work are:

1- Missing the takeaway message: The reader is expecting either a solid choice of which unsupervised loss to pick when using an adaptation method along with its hyper parameters, or a novel useful unsupervised surrogate loss function that correlates better with the performance of TTA methods. The current presentation of the paper lacks this novel aspect. Further, it was unclear (until section 6) that the model selection is done in an offline fashion. This also questions the usefulness of the conclusions in this work.

2- Clarity of the experimental setup: There are several important aspects of the experimental setup are either not clear or does not have a strong justification. Here is a list of them.
 - How is the validation done on IN-C ( page 3 ) while experiments are also carried out on IN-C (Table 2)?
 - Is this work analyzing the episodic evaluation (adaptation to one domain at a time) or a continual evaluation where different domain shifts are presented sequentially without resetting model parameters?
 - Why limiting the batch size to 10? Both Tent and EATA require a moderately large batch size (e.g. 64). I am afraid that this batch size might put such methods at a disadvantage (especially to methods employing sample rejection) making the results in this work questionable. Alternatively, one could Follow SAR with using architectures that do not use batch normalization layers (such as ViT) and test with small batch sizes.
 - If Source accuracy can be used as a cross validation metric, why not allowing RMT to leverage this information?
 - Are the model selection strategies that include unsupervised metrics (e.g. entropy) measured on the same data that the model adapted on, or there is a held out validation set where these metrics measured on?

3- Missing experiments:
- All the considered TTA methods rely on cumulative model updates while missing an important line of works in the TTA literature: data-dependent approaches such as MEMO and DDA [A, B]. Are the provided insights consistent when considering these approaches?
- Experiments with the standard batch size of 64 should be included, along with experiments with architectures that do not have batch norm layers (e.g. ViT) following the work of SAR.

4- Writing: The writing of this work can be vastly improved to make the manuscript clearer and easier to follow. Here is a small list of typos/suggestions to be considered in the updated version.
- What is "DEV" in Table 1? Shouldn't be "CON"?
- How is the perturbation $\tilde{x}_i$ in "CON" constructed?
- How is the ORACLE model selection done?
- Consider revising the presentation of Tables 2-4. For example, since all the standard deviations are relatively small, it might be worth is keeping them to the appendix.
- Regarding Figures 1-3: please consider making the x-labels, y-labels, legends, and titles larger.


[A] Memo: Test time robustness via adaptation and augmentation, NeurIPS 2022.

[B] Back to the source: Diffusion-driven test-time adaptation, CVPR 2023.

**Questions:**

In addition to the points raised in the weaknesses section, I have the following questions/suggestions:
- What is the key difference between the results presented in Table 2 and Figure 2?
- While I appreciate the experiments in Figure 3, why not following well-defined setups such as PTTA in [C]? How would the conclusions translates to methods tailored to such setups such as RoTTA [C]
- One suggestion I have regarding the writing in the experiments: please try to relate the conclusions in text with Figures and Tables. For example, Figure 2 is barely mentioned in the "No single best surrogate metric" paragraph.
- Another suggestion: try to position this work well from [D].

[C] Robust Test-Time Adaptation in Dynamic Scenarios, CVPR 2023

[D] On Pitfalls of Test-Time Adaptation, ICML 2023

---

### Official Review · Reviewer_KXPd · 2023-10-28

**Soundness:** 2 fair
**Presentation:** 3 good
**Contribution:** 2 fair
**Rating:** 3
**Confidence:** 4

**Summary:**

The paper studies test-time adaptation (TTA): where one has a model pre-trained on some source dataset, but the test data it sees is not guaranteed to be IID from the source distribution. During test time, the model needs to be adapted without any supervision/labels, only seeing test data, i.e., images from a continuous stream. There is also one important aspect: model selection, i.e., given a few models/hyper-params, one needs to choose which one will do the best without having any labeled target data for validation. The paper evaluates various TTA methods by combining them with some model selection strategies, and shows interesting patterns.

**Strengths:**

1. The paper discusses a very important question: selecting a Test-time adaptation (TTA) method in practice, based on surrogate metrics. It attempts to produce a thorough evaluation of prior methods, which can be really helpful for real-world practitioners.
2. The paper is well-motivated and easy to follow.
3. A very interesting and important finding from the results section is that using metrics that are part of the test-time adaptation procedure in model selection (e.g., using entropy when the method is Tent, or consistency loss when the method is AdaContrast) leads to bad outcomes. This is a clever insight and one that can significantly help practitioners. **I would recommend including this in the abstract**.
4. The results from section 5, i.e., performance of TTA methods break on non-standard datasets, is an important one. I agree with the authors that a lot of TTA methods seem to be overfitting with the corruption datasets due to them being the standard benchmarks in the literature for this task. An important contribution of this paper could be to highlight that effect, and include datasets with natural distribution shift as benchmarks for TTA.

**Weaknesses:**

**Overall comment**

The paper discusses evaluating TTA methods across multiple settings, and how to choose the correct method during test-time. I would argue most of the methods/model selection strategies that are discussed in the paper are not novel and/or existed before, and the paper does not have a lot of algorithmic innovation.

While this discussion unifies various prior methods and can be a valuable guideline for practitioners to choose the appropriate TTA method, there needs to be more experiments to make it a compelling paper (i.e., add MEMO [1] as a method of comparison, add WILDS-Camelyon 17 [9], WILDS-FMoW [9], ImageNet-A [7], CIFAR-10.1 [8] as dataset benchmarks). But I do feel the problem setup is very important, and adding more experiments and a bit of rewriting can make the paper much stronger.

**Abstract and Introduction**

1. The paper mentions model restarting to avoid error propagation. There has been important work in TTA, where the model adapts its parameters to only one test example at a time, and reverts back to the initial (pre-trained) weights after it has made the prediction, doing the process all over for the next test example. This is also an important setting to consider, where only one test example is available, and one cannot rely on batches of data from a stream. For example, see MEMO [1].
2. (nitpicking, not important to include in the paper) “under extremely long scenarios all existing TTA method results in degraded performance”, while this is true, the paper does not mention some recent works that helps alleviate this. E.g., MEMO [1] in the one test example at a time scenario, or MEMO + surgical FT [2] where MEMO is used in the online setting, but parameter-efficient updating helps with feature distortion/performance degradation. So the claim is outdated.
3. It would be good to cite relevant papers such as [4] as prior works that look into model selection strategies (but not for TTA setting) to motivate the problem statement.

**Section 3.2, model selection strategies in TTA**

1. While accuracy-on-the-line [3] shows correlation between source (ID) and target (OOD) accuracies, some work [4] also say source accuracy is unreliable in the face of large domain gaps. I think table 3 shows the same result. Better to cite [4] and add their observation.
2. Why not look at agreement-on-the-line [5]? This is known to be a good way of assessing performance on the target domain without having labels. For example, A-Line [6] seems to have good performance on TTA tasks. This should also be considered as a model selection method.

**Section 4.1, datasets**

1. Missing some key datasets such as ImageNet-A [7], CIFAR-10.1 [8]. It is important to consider ImageNet-A (to show TTA’s performance on adversarial examples) and CIFAR-10.1, to show TTA’s performance on CIFAR-10 examples where the shift is natural, i.e., not corruptions. Prior work such as MEMO [1] has used some of these datasets.

**Section 4.3, experimental setup**

1. The architecture suite that is used is limited in size. Only ResNext-29 and ResNet-50 are used. Since the paper’s goal is to say something rigorous about model selection strategies, it is important to try more architectures to have a comprehensive result. At least some vision-transformer architecture is required to make the results strong. I would suggest trying RVT-small [12]  or ViT-B/32 [13].
Why do the authors use SGD as an optimizer for all tasks? It is previously shown that [14] SGD often performs worse for more modern architectures. The original TENT [15] paper also claims they use SGD for ImageNet and for everything else they use Adam [16].

**Section 5, results**

1. (Table 1) It might be easier if the texts mention that each row represent one method, and each column represents one model selection strategy. When the authors say “green” represents the best number, they mean “within a row”.
2. (Different methods’ ranking under different selection strategies) The results here are not clear and hard to read. How many times does one method outperform the other, when considering all different surrogate based metrics across all datasets? If the goal is to show consistency of AdaContrast as mentioned in the introduction, a better way of presenting this might be making something similar to table 1 of [17].
3. What does the **Median** column in Table 2 and 3 represent? There is no explanation given for this in paper.
4. I assume the 4 surrogate strategies are: S-Acc, Cross-Acc, Ent and Con. If so, then the statement **“While EATA is significantly the best under the oracle selection strategy (49.99 on average) it is outperformed for example by Tent (5th method using oracle selection) when using 3 out of 4 surrogate-based metrics”** is clearly False according to the last section of Table 2: Tent > EATA on Cross-Acc and Con, but EATA > Tent when using S-Acc and Ent.
5. **(Performance of TTA methods)** This is an interesting observation, that using non-standard benchmarks breaks a lot of popular TTA methods. If the authors can evaluate TTA on more conditions of natural distribution shift, like WILDS [9], it could really strengthen the paper.

**Questions:**

> (ii) when temporal correlation between classes is changed.

1. What is temporal correlation between classes? The order in which examples from a class appear during online TTA settings?

> for testing TTA methods we utilize the standard sequence of corruption types with the 5th level of severity sequentially, without mixing the domains.

2. What are domains here? Different corruption types? If so, it should be written as “without mixing images from different corruption types” to avoid confusion.

> Some of the TTA methods use data augmentation at test-time (Wang et al., 2022; Sun et al., 2020), including AdaContrast and RMT considered in this paper. Those augmentations include some of those that were used to generate ImageNet-C and CIFAR100-C benchmarks (details in the Appendix)

3. Is it possible to use different augmentations like AugMix [10] or RandMix [11] for these methods instead of using corruptions similar to those in ImageNet-C/CIFAR-100-C? Also how about holding up some corruptions as “validation corruptions” and use them to generate the augmentations? **(This is not a limitation of this paper, rather I want to clarify the methodology from one of the baselines this paper uses)**

4. How is source accuracy used with TENT, for example? Do you run TENT, with a certain set of hyper-params, on a heldout source dataset, look at the accuracy, and choose the hyper-params that lead to the highest accuracy? This point / similar points maybe clarified with an example in the paper.

[1] MEMO: Test Time Robustness via Adaptation and Augmentation, https://arxiv.org/abs/2110.09506

[2] Surgical Fine-Tuning Improves Adaptation to Distribution Shifts, https://arxiv.org/abs/2210.11466

[3] Accuracy on the Line: on the Strong Correlation Between Out-of-Distribution and In-Distribution Generalization, https://proceedings.mlr.press/v139/miller21b.html

[4] Towards Accurate Model Selection in Deep Unsupervised Domain Adaptation, https://proceedings.mlr.press/v97/you19a.html

[5] Agreement-on-the-Line: Predicting the Performance of Neural Networks under Distribution Shift, https://arxiv.org/abs/2206.13089

[6] Reliable Test-Time Adaptation via Agreement-on-the-Line, https://arxiv.org/abs/2310.04941

[7] Natural Adversarial Examples, https://arxiv.org/abs/1907.07174v4

[8] Do CIFAR-10 Classifiers Generalize to CIFAR-10? https://arxiv.org/abs/1806.00451

[9] WILDS: A Benchmark of in-the-Wild Distribution Shifts, https://arxiv.org/abs/2012.07421

[10] AugMix: A Simple Data Processing Method to Improve Robustness and Uncertainty, https://arxiv.org/abs/1912.02781

[11] RandomMix: A mixed sample data augmentation method with multiple mixed modes, https://arxiv.org/abs/2205.08728

[12] Towards robust vision transformer, https://arxiv.org/abs/2105.07926

[13] Learning Transferable Visual Models From Natural Language Supervision, https://arxiv.org/abs/2103.00020

[14] How to Fine-Tune Vision Models with SGD, https://arxiv.org/abs/2211.09359

[15] Tent: Fully Test-Time Adaptation by Entropy Minimization, https://openreview.net/forum?id=uXl3bZLkr3c

[16] Adam: A Method for Stochastic Optimization, https://arxiv.org/abs/1412.6980

[17] No True State-of-the-Art? OOD Detection Methods are Inconsistent across Datasets, https://arxiv.org/abs/2109.05554

---

### Official Review · Reviewer_tHzX · 2023-10-31

**Soundness:** 2 fair
**Presentation:** 2 fair
**Contribution:** 1 poor
**Rating:** 3
**Confidence:** 4

**Summary:**

This paper offers an empirical investigation into the problem of model selection for test-time adaptation (TTA). The study encompasses the evaluation of six TTA approaches using six distinct model selection strategies across four TTA datasets. The empirical study yields several findings: (1) The accuracy of model selection strategies exhibits significant variation across TTA datasets and methods. (2) Among the TTA methods evaluated, only AdaContrast proves capable of consistently achieving surrogate-based model selection that aligns with the oracle selection. (3) Model selection for TTA proves to be a challenging task, and the use of a small labeled set stands out as an effective validation baseline.

**Strengths:**

1. This paper delves into the significant issue of unsupervised model selection in the presence of distribution shifts, a topic of great importance that has received limited exploration.

2. Extensive experiments are conducted to assess the performance of existing model selection methods when applied to state-of-the-art TTA approaches.

3. The findings obtained offer valuable insights for guiding future research in the field of TTA.

**Weaknesses:**

1. This paper lacks novelty as it shares considerable overlap with existing works. In the broader context of model selection under distribution shifts, a comprehensive empirical study on model selection for unsupervised domain adaptation has been conducted [1]. In a narrower scope, recent publications on TTA [2, 3] have also provided extensive empirical studies on model selection in this context. In comparison to these relevant studies, this paper may not introduce sufficiently novel insights into the model selection problem under distribution shifts. Specifically, the second finding pertains to the characteristics of a specific TTA method, AdaContrast. Therefore, the first and third findings, which are valuable and more general, appear to have already been introduced in [1, 3].

2. This paper only investigates the offline model selection, which may have limited insights to address the challenge of online model selection for TTA methods, particularly in light of the batch dependency analyzed in [3].

3. The empirical study in this paper lacks comprehensiveness. In terms of model selection baselines, only six surrogate-based selection metrics are considered, with some key metrics introduced in [1] missing. Regarding the TTA methods, fewer methods are considered compared to a relevant study [3], which encompasses eleven TTA methods with varying methodologies.

4. The presentation of the paper lacks clarity and organization, which is evident in the significant disorder between tables/figures and the textual analysis. This lack of organization substantially reduces the overall readability of the paper.

References

[1] Benchmarking Validation Methods for Unsupervised Domain Adaptation. arxiv 2022

[2] Parameter-free Online Test-time Adaptation. cvpr 2022

[3] On Pitfalls of Test-Time Adaptation. icml 2023

**Questions:**

The primary concerns are outlined in "Weaknesses".

**Details Of Ethics Concerns:**

Dear ACs and ethics reviewers,

I observed **a possible disclosure of authors** in the attached Supplementary Material while reviewing the provided code. Specifically, you can refer to the "README.md," where you will find the statement, "This is an open source online test-time adaptation repository based on PyTorch. It is joint work by Robert A. Marsden and Mario Döbler."

Thank you for your attention.

---

### Official Review · Reviewer_34fr · 2023-11-02

**Soundness:** 2 fair
**Presentation:** 2 fair
**Contribution:** 2 fair
**Rating:** 5
**Confidence:** 4

**Summary:**

The paper presents a comprehensive analysis of model selection strategies for TTA. Specifically, it examines six model selection techniques applied to six recent TTA methods. The empirical analyses lead to three key observations:
  1. Notable variations in the effectiveness of the model selection techniques across different datasets and TTA methods.
  2. Except for AdaContrast, the outcomes of TTA methods using surrogate model selection are much worse than their oracle model selection counterparts.
  3. Leveraging a small number of labeled test samples can greatly help model selection.

**Strengths:**

Model selection is a paramount open issue for TTA. While recent studies have emphasized its significance, a concrete solution has been elusive. This paper makes a great attempt to explore potential options, presenting an in-depth analysis of their properties and limitations.

**Weaknesses:**

Despite extensive efforts to evaluate six model selection methods, the choice of these methods seems questionable.
- Most of them have obvious flaws in their design, e.g., using the adaptation signal (entropy of class prediction, consistency under augmentation). As such, the limitations of these surrogate methods are somewhat predictable. It's not very clear what insights this study brings up and how they may help design a better model selection method
- Model selection techniques have been studied for unsupervised domain adaptation (UDA), and some could be either applied or tailored to TTA. Incorporating these techniques in the benchmark may greatly enrich the analysis.

[1] Towards Accurate Model Selection in Deep Unsupervised Domain Adaptation, ICML'19
[2] Unsupervised Validation of Domain Adaptation via Soft Neighborhood Density, ICCV'21

**Questions:**

Why does the model selection method based on source accuracy work well for AdaContrast but not others?
A deeper dive into this finding would be interesting.